

# Evaluation of structural and ultrastructural changes in thyroid and parathyroid glands after near infrared irradiation: study on an animal model

Carlos Serra[1,2]   Luis Silveira[2]

[1] Hospital do SAMS, Lisboa, Portugal
[2] Universidade da Beira Interior, Faculdade de Ciências da Saúde, Covilhã, Portugal

## ABSTRACT

Intraoperative identification of parathyroid glands is a tough task for surgeons performing thyroid or parathyroid surgery, because the small size, color and shape of these glands hinder their discrimination from other cervical tissues. In 2011, Paras described the autofluorescence of parathyroid glands, a property that could facilitate their intraoperative identification. Parathyroid glands submitted to a 785 nm laser beam emit fluorescence in the near infrared range, with a peak at 822 nm. As the intrinsic properties of secretory tissues may be affected by the exposure to the near infrared light, a situation that could preclude their intraoperative utilization, the authors compared the structural and ultra-structural patterns of rat's thyroid and parathyroid glands submitted to irradiation replicating the conditions that allow their intraoperative identification, with those of non irradiated animals. Twenty-four Wistar rats were divided into six groups: animals of Groups 1, 3 and 5 were submitted under general anesthesia to direct irradiation of the cervical area with a 780 nm LED light for 3 minutes through a cervical incision, and animals of Groups 2, 4 and 6 were submitted to cervical dissection without irradiation. Animals of were euthanized immediately (Groups 1 and 2), at Day 30 (Groups 3 and 4) at and at Day 60 (Groups 5 and 6) and thyroid and parathyroid glands were removed: one lobe was prepared for conventional pathological examination and the other lobe for electron microscopy observed by three experienced pathological experts. Twenty-four samples were prepared for conventional histology and there were no alterations reported in any group. Due to technical problems, only 21 samples were observed by electron microscopy and there were no differences in the ultrastructure of parathyroid and thyroid glands, namely the nuclear pattern, mitochondria, endoplasmic reticulum or secretory granules, in any of the groups. These results confirm the innocuity of near infrared irradiation', allowing its intraoperative utilization.

# INTRODUCTION

Hypoparathyroidism is a frequent complication of thyroid surgery, occurring in up to 30% of total thyroidectomies, and is an important cause of morbidity and litigation.

Corresponding author
Carlos Serra, caaserra@netcabo.pt, caaserra@me.com

The parathyroid glands' small size, colour and resemblance to lymph nodes and fat tissue make their intraoperative identification difficult and highly dependent on the surgeon's experience (*Akerstrom, Malmaeus & Bergstrom, 1984*; *Bergenfelz et al., 2020*; *Edafe & Balasubramanian, 2017*). The development of an effective optical device to help surgeons in the identification of parathyroid glands in thyroid and parathyroid surgery could have a strong impact on the reduction of postoperative hypocalcaemia.

*Paras et al. (2011)* described the autofluorescence of parathyroid glands when exposed to a near-infrared light source, a property that allows their discrimination from other cervical tissues. Parathyroid glands submitted to a 785 nm laser beam emit fluorescence with a peak at 822 nm. Even though the thyroid gland also emitted fluorescence at the same wavelength, its inferior intensity allows easy discrimination.

Since that first description, other groups have confirmed these results (*DeLeeuw et al., 2016*; *Falco et al., 2016*; *Kim et al., 2016*; *McWade et al., 2013*). Near-infrared radiation is characterized by its capacity to penetrate up to five mm into biological tissues (*Lakowicz, 2006*; *Monici, 2005*; *Stolik et al., 2000*). Although the light intensity used for stimulation is small, it is not clear whether it is harmless for tissues; therefore, it is important to test the response of the secretory tissues to the irradiation by studying possible structural and ultrastructural changes as well as changes in the secretory pattern.

In the literature, works on the effects of utilization of near-infrared radiation for diagnostics and localization of thyroid and parathyroid glands are scarce; however, there are some publications on the effects of this radiation when used with therapeutic intention in dental medicine (*Azevedo et al, 2005*; *Fronza et al., 2013*; *Lerma et al., 1991*; *Mayer et al., 2013*; *Mohammed, Al-Azawi & AL-Mustawfi, 2011*; *Weber et al., 2014*).

In spite of the lack of information about possible effects on the structure or function of thyroid and parathyroid glands submitted to near-infrared irradiation, several works have stated its clinical application for intraoperative identification of parathyroid glands with relevant impact in clinical practice (*Dip et al., 2019*; *Kim et al., 2016*; *Ladurner et al., 2017*; *McWade et al., 2016*).

In a previous work, the authors studied the influence of near-infrared radiation emitted by a 780 nm LED light on the secretory pattern of thyroid and parathyroid glands, replicating the conditions that allow the identification of parathyroid glands by autofluorescence, and concluded that although some alterations occurred, they are transient and without great significance, thus confirming the safety of autofluorescence for this purpose (*Serra & Silveira, 2019*).

Weber showed that utilization of a low-intensity laser applied to dental alveola of rats after dental extraction and implant collocation could influence thyroid function and calcium levels, whereas Fronza did not find any changes in thyroid hormone levels in rabbits using GaAlAs lasers ($\lambda = 830$ nm, 120 J/cm$^2$) (*Fronza et al., 2013*; *Weber et al., 2014*).

Lerma studied structural and ultrastructural changes in thyroid glands of rats submitted to HeNe lasers ($\lambda = 632$ nm, 75 J/cm$^2$) and found mild degenerative changes on conventional histology that recover in less than 3 months. Some minor long-term lesions

were also identified using electron microscopy: an increased number of peroxisomes and crystalline structures (*Lerma et al., 1991*).

In all these studies, exposure of the thyroid and parathyroid area was not direct because irradiation was done via the skin and adjacent soft tissue and only one study analysed possible structural or ultrastructural modifications in the tissues submitted to near-infrared radiation. Furthermore, the different characteristics of the radiation used preclude comparisons.

The structural unit of the thyroid gland in rats, as in humans, is the follicle, which is filled with a secretory product called colloid. Follicles are irregularly spheroid and separated by a delicate fibrovascular stroma containing abundant capillaries, lymphatics and nerves (*Asa & Mete, 2018*).

Human parathyroid glands (usually four) are tiny yellow-brown glands of size 6 × 3 mm localized posteriorly to the thyroid gland (*Akerstrom, Malmaeus & Bergstrom, 1984*; *Policeni, Smoker & Reede, 2012*). By contrast, rats only have one pair of parathyroid glands (originating from the third pharyngeal pouch, as with the human inferior parathyroid glands) that migrate caudally, accompanying the caudal migration of the thymus (*Isono, Shoumura & Emura, 1990*; *Kittel, Ernst & Kamino, 1996*).

Parathyroid glands are surrounded by a thin fibrous capsule that covers a net of fat tissue, vessels and glandular parenchyma. A rich network of arterioles, venules and capillary vessels vascularize the glands (*Chen, Emura & Kubo, 2013*; *Isono, Shoumura & Emura, 1990*; *Kittel, Ernst & Kamino, 1996*).

Normal human parathyroid glands present two types of cells:

- chief cells—small dark cells that produce and release parathyroid hormone (PTH) and turn clear when they release PTH;
- oxyphilic cells—with clear tonality and unknown function, whose number increases with age (*Asa & Mete, 2018*).

Rat parathyroid glands are composed only of chief cells. Different tones of chief cells (dark or lighter), which are associated with different functional status, can be seen in adult rats (*Isono, Shoumura & Emura, 1990*; *Kittel, Ernst & Kamino, 1996*). Chief cells in rats are polygon shaped and organized in cordons or laminate surrounded by a thin stroma of connective tissue and a capillary net (*Isono, Shoumura & Emura, 1990*; *Kittel, Ernst & Kamino, 1996*). Using conventional staining, the cytoplasm appears granular and is surrounded by an almost imperceptible cellular membrane. Nuclei are round or oval with a fine dispersion of chromatin (*Kittel, Ernst & Kamino, 1996*).

On transmission electron microscopy, the plasmatic membrane of chief cells in rats is smooth on the basal side; on the lateral sides, the cells are connected by desmosomes or tight junctions, whereas smooth areas or interdigitations between cells can be found in the internal portion of the cordons. Nuclei have a basal localization and are often indented, with medial electronic density (*Kittel, Ernst & Kamino, 1996*). The Golgi apparatus is well developed and located on the apical portion of the cell, as are the secretory granules (*Isono, Shoumura & Emura, 1990*; *Kittel, Ernst & Kamino, 1996*).

Two types of secretory granules can be found (*Chen, Emura & Kubo, 2013*; *Isono, Shoumura & Emura, 1990*; *Kittel, Ernst & Kamino, 1996*):

- small granules with a large halo (100–400 nm);
- large granules with a smaller halo (500–700 nm).

Mitochondria, rough endoplasmic reticulum and free ribosomes can be found in the chief cells. Rat mitochondria are long and wavy structures (*Kittel, Ernst & Kamino, 1996*).

In this work we evaluate the structural and ultrastructural patterns of thyroid and parathyroid glands submitted to a 780 nm LED light—replicating the conditions that allow their identification by autofluorescence in an animal model—compared with those of non-irradiated animals.

## MATERIALS & METHODS

Twenty-four male Wistar rats aged 10–12 weeks, produced in the vivarium of the School of Health Sciences of the University of Beira Interior (UBI), were used in the study. The experimental protocol was approved by the General Directorate of Agriculture and Veterinary Science (Approval number 0421/000/000/2018).

Experiments were conducted according to the methodology described by the same authors is a previous work namely animal housing, anesthesia, surgical procedure and irradiation conditions (*Serra & Silveira, 2019*).

The rats were kept in individual ventilated cages (Sealsafe Plus GR 900; Tecniplast S.p.A., Bugiggiatte, Italy) in an animal room under controlled conditions (temperature: 22–24 °C; relative humidity: 40–60%) and provided with food (Standard Diet 4RF18; Mucedola srl, Settimo Millanese, Italy) and water ad libitum.

Rats were divided into six groups (three experimental; three control) of four animals each and submitted to general anaesthesia via isoflurane inhalation. General anaesthesia was induced with 5% isoflurane (IsoFlo; Zoetis Inc., Parsippany, NJ, USA) in oxygen (2.4 l/min) and maintained with 2% isoflurane in oxygen (2.4 l/min) delivered by a Matrx VME 2N anaesthesia machine (Matrx Inc., Orchard Road, New York, USA). After a horizontal cervical incision, neck tissues were dissected to expose the trachea and identify the thyroid gland.

For animals in Groups 1, 3 and 5, a 780 nm light beam emitted by a collimated LED (M780 L3C1; Thorlabs, Newton, NJ, USA) with a 780 nm excitation bandpass filter (84121 Edmund Optics, Barrington, NJ, USA) was delivered to the cervical area for 180 s (dose: 1.37 J/cm$^2$). The fluorescence emitted by the parathyroid and thyroid glands was visualized with a Nikon D70 DSLR camera (Nikon Corporation, Tokyo, Japan) converted to infrared radiation capture and a Nikon Nikkor 50 mm f/1.8 AF M/A lens (Nikon Corporation, Tokyo, Japan) with an 832 nm emission bandpass filter (84123 Edmund Optics, Barrington, NJ, USA). To ensure equal exposure distance for all animals, the LED was coupled to a retention device (Fig. 1), with the focus placed approximately 20 cm from the cervical region (irradiated area: 20 mm; power density: 0.067 W/cm$^2$). The exposure time was calculated based on the times reported in the literature for identification of parathyroid

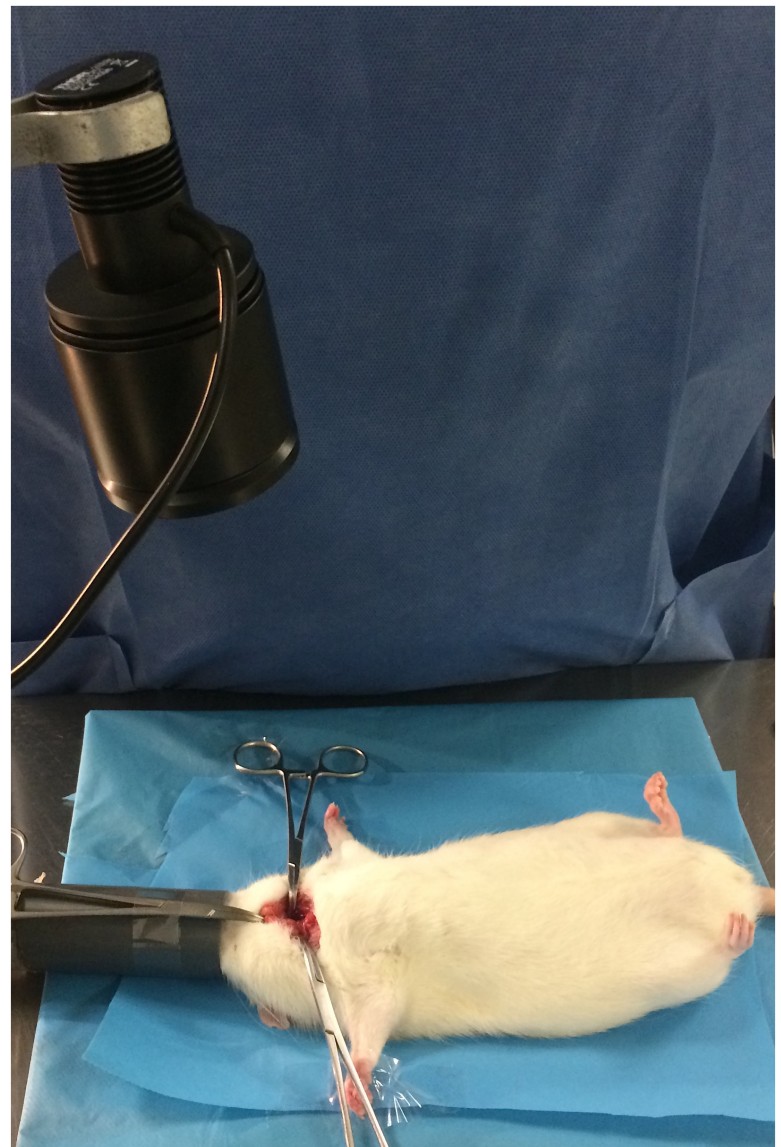

**Figure 1  Intraoperative image showing the LED source.**

glands using similar systems (*Kim et al., 2018*). For animals in Groups 2, 4 and 6, the procedure was done without neck infrared irradiation.

Rats in Groups 1 and 2 were euthanized immediately after surgery and a thyroparathyroidectomy was performed. Euthanasia was performed by cardiac puncture under general anaesthesia. Thyroparathyroid blocks were divided by the isthmus, with one half fixed in 10% formalin for conventional histological examination and the other half fixed in gluteraldehyde for transmission electron microscopy, both conserved at a temperature of 5 °C. Animals in Groups 3, 4, 5 and 6 were maintained in similar conditions after surgery, medicated with an analgesic agent (meloxicam 1 mg/kg sc, Maxicam 0.2%; Ourofino Saúde Animal, Osasco, Brazil) and submitted to the same procedure at Day 30 (Groups 3

and 4) and Day 60 (Groups 5 and 6) postoperatively. Collected tissues were prepared for conventional histological examination at the Centro de Diagnóstico Anatomopatológico (Lisbon, Portugal), stained with haematoxylin-eosin and lamel mounted.

Preparation for electron microscopy was done at the Pathology Department of the Hospital Professor Doutor Fernando Fonseca (Amadora, Portugal) using the following procedure: after collection, samples were fixed in gluteraldehyde and buffered in 0.1 M sodium cacodylate with sucrose (primary fixer), 1% osmium tetroxide in 0.1 M sodium cacodylate with sucrose (secondary fixer) and 0.5% uranyl acetate in 0.1 M acetate acetic acid. Samples were then dehydrated with crescent concentrations of ethyl alcohol, transitioned with propylene oxide and then embedded in epoxy resin (araldite). Ultrathin cuts were made with a Leica EM UC7 RT ultramicrometer in copper grills (200 mesh, 79 nm) and contrasted using Reynolds lead citrate.

Three experienced pathologists examined the samples using optical microscopy, searching for morphological changes in the thyroid or parathyroid glands without knowing about the exposure to near-infrared radiation. A brief report was given of each observation. Observations with transmission electron microscopy were made with a Hitachi H-7650 120 kV transmission electron microscope in the Laboratório de Microscopia Electrónica, Instituto Gulbenkian de Ciência. Selected images were observed by the same pathologists who carried out the histological examination.

## RESULTS

Table 1 shows the results of optical microscopy of samples collected immediately, 30 days and 60 days after exposure, respectively. Pathological examination of the samples did not find significant alterations to the usual follicular structure of thyroid cells in any of the groups. Structural changes also were not identified in the parathyroid glands, with the chief cells in all the groups having the same characteristics.

Technical problems occurred during processing of the tissues so that only 21 samples were amenable to observation using electron microscopy. Table 2 shows the electron microscopy results for samples collected immediately, 30 days and 60 days after exposure. No qualitative ultrastructural alterations were observed using electron microscopy in either the exposed or control groups, suggesting that there was no degeneration of any organelles: mitochondria, rough endoplasmic reticulum, secretory granules and nucleus of both groups had the same characteristics.

The presence of peroxisomes or crystalline structures was not observed in any sample. As with conventional histology, no ultrastructural features allowed the distinction between irradiated and non-irradiated tissues using electron microscopy. Figures 2 and 3 show the optical microscopy and electron microscopy images acquired for samples of Groups 1 and 2.

## DISCUSSION

The possibility that irradiation of the thyroid and parathyroid area could promote structural or ultrastructural changes in targeted tissues must be considered when facing

**Table 1  Results of optical microscopy of samples collected immediately after exposure, 30 and 60 days after exposure.**

|  | $n$ | Thyroid identified | Morphological alterations on thyroid | Parathyroid identified | Morphological alterations on parathyroid |
|---|---|---|---|---|---|
| **After exposure** | | | | | |
| Exposed (group 1) | 4 | 4 | 0 | 4 | 0 |
| Control (group 2) | 4 | 4 | 0 | 3 | 0 |
| **30 days after exposure** | | | | | |
| Exposed (group 3) | 4 | 4 | 0 | 4 | 0 |
| Control (group 4) | 4 | 4 | 0 | 3 | 0 |
| **60 days after exposure** | | | | | |
| Exposed (group 5) | 4 | 4 | 0 | 4 | 0 |
| Control (group 6) | 4 | 4 | 0 | 3 | 0 |

**Table 2  Results of electron microscopy of samples collected immediately after exposure, 30 and 60 days after exposure.**

|  | $N$ | Ultrastructural alterations | |
|---|---|---|---|
|  |  | Thyroid | Parathyroid |
| **After exposure** | | | |
| Exposed (group 1) | 4 | 0 | 0 |
| Control (group 2) | 3 | 0 | 0 |
| **30 days after exposure** | | | |
| Exposed (group 3) | 4 | 0 | 0 |
| Control (group 4) | 3 | 0 | 0 |
| **60 days after exposure** | | | |
| Exposed (group 5) | 4 | 0 | 0 |
| Control (group 6) | 3 | 0 | 0 |

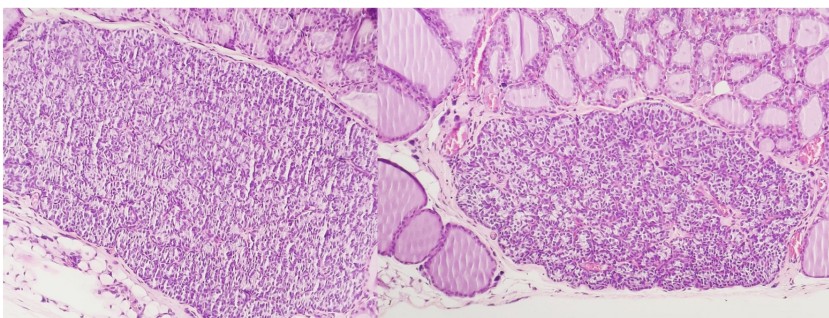

**Figure 2  Optical microscopy of parathyroid samples of groups 1 (left) and 2 (right) HE X100.**

the known tissue penetration capacity of near-infrared radiation allied to intrinsic features of endocrine tissues (*Stolik et al., 2000*). As the magnitude of these changes could preclude the clinical application of near-infrared autofluorescence for intraoperative identification

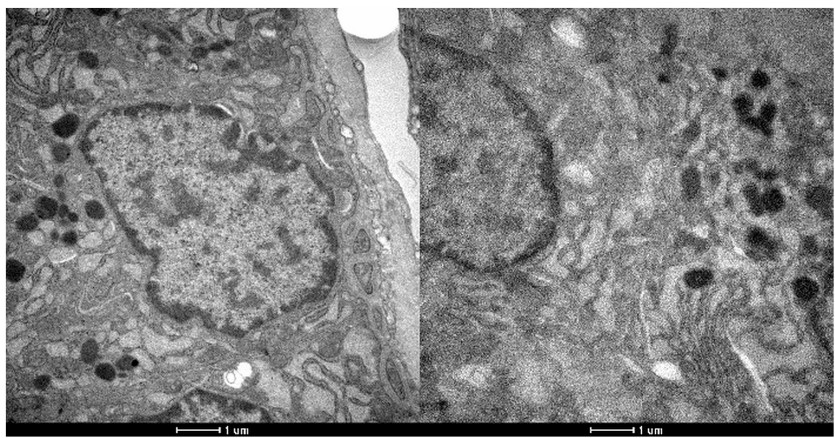

**Figure 3 Electron microscopy of parathyroid samples of groups 1 (left) and 2 (right).** X 25600.

of parathyroid glands, this work reinforces the safety of exposure in a controlled setting, as suggested in a previous study by the same authors directed to changes in secretory pattern and weight variation, thus allowing its utilization (*Serra & Silveira, 2019*).

As the doses used were above the maximum necessary for intraoperative identification of parathyroid glands in the rat model, these results can be extrapolated for utilization of this technique in human surgery, in spite of the absence of oxyphilic cells in rats. In our sample no degenerative changes were found in the thyroid gland's conventional histology and no significant difference was documented regarding the presence of peroxisomes or crystalline structures on electron microscopy. Special attention was paid to the features in parathyroid mitochondria and secretory granules but no differences were found between irradiated and non-irradiated tissues. Furthermore, no alterations were found in the nuclei or rough endoplasmic reticulum.

Our results are aligned with the work of Lerma , where no significant structural or ultrastructural changes could be found in the thyroid glands of rats submitted to a 635 nm laser beam (*Lerma et al., 1991*). More important and persistent alterations could be associated with excessive exposure to light for consecutive days, as shown by Parrado (*Parrado et al., 1999*), who found a direct relationship between the severity of the injury and the energy of the applied irradiation. This excessive exposure may also explain the significant alteration in hormonal level but no change in the histological structure of the tissues found by Azevedo between the first and seventh day after the last irradiation (*Azevedo et al., 2005*). The deeper penetration of the laser compared with the LED for the same wavelength could also help to explain the alterations found.

In spite of the absence of structural or ultrastructural alterations in the parathyroid and thyroid glands provoked by exposure to near-infrared irradiation, this study did not evaluate possible changes in the secretory pattern of the glands, which is a major limitation. However, in a previous work the authors showed, in another set of animals, that mild and transient alterations can occur without any clinical impact (*Serra & Silveira, 2019*).

## CONCLUSIONS

The absence of morphological changes in thyroid and parathyroid glands coupled with the mild and transient alterations in secretory pattern of the glands submitted to near-infrared irradiation reinforces the hypothesis that the utilization of near-infrared light for intraoperative identification of parathyroid glands is harmless for the tissues and can be used safely in clinical practice. However, further investigation should be performed to give robustness to this study.

## ACKNOWLEDGEMENTS

The authors want to thank Drs. Paula Guerra Marques, Sara Turpin and Delfina Brito from the Pathology Department of Hospital dos Sams, for their invaluable help on both optical and electron microscopy observations. We are also grateful to Professor Erin Tranfield from Instituto Gulbenkian de Ciência that made preliminary electron microscopy of the samples and select the images and to Dra. Rita Manso from Hospital Fernando Fonseca that prepared the samples for observation. For the performance of animal surgeries, we had the invaluable collaboration of technicians Maria José and Maria João from Universidade da Beira Interior and Dra. Maria do Rosário Custódio, Master on animal behavior.

### Funding

The authors received no funding for this work.

### Competing Interests

The authors declare there are no competing interests.

### Author Contributions

- Carlos Serra and Luis Silveira conceived and designed the experiments, performed the experiments, analyzed the data, prepared figures and/or tables, authored or reviewed drafts of the paper, and approved the final draft.

### Animal Ethics

The following information was supplied relating to ethical approvals (i.e., approving body and any reference numbers):

Direcção Geral de Agricultura e Veterinária provided full appoval for this research (0421/000/000/2018).

### Data Availability

The summary reports of the optic microscopy are available in the Supplementary File. The number of analysed samples and occurrence of pathological changes, without use of any scores are available in Tables 1 and 2.

## Supplemental Information

Supplemental information for this article can be found online at http://dx.doi.org/10.7717/peerj.11891#supplemental-information.

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
