# Peer review of "Evaluation of structural and ultrastructural changes in thyroid and parathyroid glands after near infrared irradiation: study on an animal model"

_PeerJ, doi:10.7717/peerj.11891_

## Round 0.1 · original submission · Major Revisions

I agree with all three reviewers that there is considerable room for improvement in the linguistic aspects. Using terms like "couldn't" is also too informal, for example.

A major limitation, as the third reviewer points out, is the lack of investigation into whether NIR impacts thyroid/parathyroid function (in terms of amounts of secreted hormones) or even the biochemical functionality of secreted hormones.

·

Basic reporting

English language should be revised in some sections of the article, mainly: Seldom the word “on” is used when “in” should be applied, some phrases are long and unclear, the word “also” is inappropriately used in some phrases.
Abstract: row 24, 26-30, 35-38, 42
Intro: 54, 66-71, 77-78, 84-88, 91
Materials and methods: 178 there is a double “buffered in”
Results: 203 the word respectively should be dropped
Discussion: 220-224
Conclusions: 240-241
Intro & background to show context.
The introduction and background of the paper are adequate. The authors present the issue with great detail and accuracy. Row 69-73 state: “In the literature, works on the effects of utilization of near infrared radiation for diagnostics and localization of thyroid and parathyroid glands are scarce, existing however some publications about the effects of this radiation when used with therapeutic intention on dental medicine”. In my opinion, this phrase is misleading. There are scars to nill literature on the effects of NIR on the structure and function of tissues. However, several papers are dealing with the use of this method of parathyroid identification in the clinical setting and its impact on clinical practice. I think that this should be highlighted. (kim et al. surgical oncology 2020, Solorzano et al. Surgery 2021, etc)
Literature well referenced & relevant. - yes
Structure conforms to PeerJ standards, discipline norm, or improved for clarity. - yes
Figures are relevant, high quality, well labelled & described. - yes

Experimental design

Original primary research within scope of the journal.
In my opinion, the papers fit well with the scope of the journal.
Research question well defined, relevant & meaningful.
The study investigates the impact of NIR on parathyroid and thyroid tissue to assess its safety. Even though this method of parathyroid identification is FDA approved and in current use, I didn’t find articles analyzing the possible impact of NIR on parathyroid tissue. It makes this paper original and relevant.
It is stated how the research fills an identified knowledge gap.
As pointed out above, the knowledge gap filed with this study should be more accurately presented, and the fact that there is scarce literature about structural changes with the use of NIR highlighted.
Rigorous investigation performed to a high technical & ethical standard. - yes
Methods described with sufficient detail & information to replicate.
There should be a reference or an explanation after row 160-161 (times of exposure).

Validity of the findings

Meaningful replication encouraged where rationale & benefit to literature is clearly stated.
The paper cites Lerma et al. regarding thyroid structural changes. The authors found minor temporary changes on conventional histology and some minor persistent changes on electron microscopy. In the current study, no structural changes were found. Although the methods varied between the two experiments, the authors should encourage further investigation.
All underlying data have been provided; they are robust, statistically sound, & controlled.

Additional comments

I am honored to be granted the opportunity to review this article. I congratulate the authors on a fascinating and inspiring study. The topic of the investigation is original, and the literature on the matter is scarce. The paper lacks major issues. On my account, English should be revised as above. Some minor issues are present and should be addressed.

Reviewer 2 ·

Basic reporting

Scientific work must be formal and impersonal. Because of this, the construction of the sentence in the first or third person must be avoided.
Discussion - lines 232, 234
“We dedicated special attention to features on parathyroid mitochondria and secretory granules, without achieving any differences between irradiated and non-irradiated tissues.
We also couldn’t find nuclear or rough endoplasmic reticule alterations.”

Experimental design

It would be interesting to mention the power density of the LED that was used and the irradiated area, as the focus was approximately 20 cm from the cervical region.
A figure could be added in the text.

Validity of the findings

The discussion is very short and was not explored. It would be interesting to include that a possible alteration of the glands may be associated with the excessive use of light for consecutive days, as we can see in the study by Parrado et al (1989). The glands were irradiated for 15 consecutive days, for example. Evaluation of changes in the densities of epithelial, colloidal and follicular volumes and the activation index revealed that the laser produces changes in the thyroid parenchyma. It was noted that there is a direct relationship between the severity of the injury and the energy of the applied irradiation.
Azevedo et al (2005) - A statistically significant hormonal level alteration between the first day and 7 days after the last irradiation was found, but no changes were found in the histological structure of the glands.
It is important to remember that the penetration of the laser is deeper than LED, even if it is the same wavelength. The LED is neither collimated nor coherent.

Additional comments

It is a very interesting study, but it could be better analyzed and explored in the discussion

Reviewer 3 ·

Basic reporting

The article should be reviewed by an English speaker for proper grammar. For instance, the authors should avoid contractions such as "couldn’t" and should use "could not" instead.

The literature references provide sufficient background and context to the studies.

Experimental design

The experimental design is good and the research question is well defined.

The numbers of rats in the study are adequate.

Validity of the findings

The finding show no structural differences in the parathyroid and thyroid glands of mice undergoing near infrared exposure for autofluorescence versus control mice.

However, the authors should include a paragraph in the Discussion section regarding the limitations of the study. For instance, although there were no structural differences between groups, the authors did not assess thyroid or parathyroid function in the rats and therefore cannot comment on the the possibility of functional alterations of the glands from radiation exposure.

---

## Round 0.2 · accepted · Accept

Thank you for submitting this paper to PeerJ.

Reviewer 2 ·

Basic reporting

Self-contained with relevant results to hypotheses.

Experimental design

Methods described with sufficient detail and information to replicate

Validity of the findings

Conclusions are well stated, linked to original research question and limited to supporting results

Additional comments

I would like to congratulate you for reviewing the text. This is a study of great relevance.